# Cholinergic Chemotransmission and Anesthetic Drug Effects at the Carotid Bodies

**DOI:** 10.3390/molecules25245974

**Published:** 2020-12-17

**Authors:** Maarten Honing, Chris Martini, Monique van Velzen, Marieke Niesters, Albert Dahan, Martijn Boon

**Affiliations:** Department of Anesthesiology, Leiden University Medical Center, 2333 ZA Leiden, The Netherlands; g.h.m.honing@lumc.nl (M.H.); C.H.Martini@lumc.nl (C.M.); M.van_Velzen@lumc.nl (M.v.V.); m.niesters@lumc.nl (M.N.); a.dahan@lumc.nl (A.D.)

**Keywords:** peripheral chemosensitivity, anesthesia, muscle relaxants, cholinergic transmission

## Abstract

General anesthesia is obtained by administration of potent hypnotics, analgesics and muscle relaxants. Apart from their intended effects (loss of consciousness, pain relief and muscle relaxation), these agents profoundly affect the control of breathing, in part by an effect within the peripheral chemoreflex loop that originates at the carotid bodies. This review assesses the role of cholinergic chemotransmission in the peripheral chemoreflex loop and the mechanisms through which muscle relaxants and hypnotics interfere with peripheral chemosensitivity. Additionally, consequences for clinical practice are discussed.

## 1. Introduction

General anesthesia is a pharmacologically induced state which allows exposure to surgical trauma without memory or harmful sympathetic activation. The anesthetic state is typically obtained by administration of potent hypnotics and opioid analgesics. In addition, pharmacologic paralysis of skeletal muscles facilitates endotracheal intubation and optimizes surgical working conditions. Apart from their intended effects, anesthetic agents also disrupt the control of breathing through effects at central and peripheral chemoreceptive sites [1,2,3,4,5]. Peripheral control of breathing is mediated through the carotid bodies, which are sensitive to derangements in arterial oxygen, carbon dioxide and glucose levels [6]. Especially muscle relaxants and intravenous and inhalational anesthetics depress peripheral chemosensitivity [1,4,5,7,8,9]. Attenuated peripheral chemosensitivity following surgery, when the patient is just off artificial ventilation, may precipitate adverse respiratory events in the postoperative period. The mechanism by which muscle relaxants and the intravenous anesthetic agent propofol influence peripheral chemosensitivity involves disruption of cholinergic chemotransmission at the carotid bodies. In this review, cholinergic chemotransmission at the carotid bodies is reviewed and possible mechanisms of anesthetic interference with cholinergic chemotransmission are discussed.

## 2. Chemosensing at the Carotid Bodies

All mammalian cells require oxygen to maintain aerobic homeostasis. Failure to maintain aerobic metabolism is a potentially lethal condition that should be prevented at all costs. Therefore, mammalian organisms possess chemoreceptive tissue, aimed at detecting and overcoming detrimental effects of hypoxia and other homeostatic derangements (e.g., hypercapnia, hypoglycemia, acidosis and hyperthermia). There are two sets of chemoreceptors that play a pivotal role in detecting and responding to these perturbations: the peripheral and central chemoreceptors. The latter are located in the brainstem of the central nervous system and react primarily to changes in arterial CO_2_ and acidosis. Peripheral chemoreceptive tissue can be found at the aortic arch, the aortic bodies, and at the carotid bifurcation, the carotid bodies, which are considered the primary peripheral chemoreceptors. The carotid bodies are highly vascular organs that are strategically located at the bifurcation of the common carotid artery. They consist of two cell types (i.e., type 1 and type 2 cells) of which type 1 (or glomus) cells are the primary chemosensors and type 2 cells are supportive or sustentacular cells (very much alike glia cells in the central nervous system) [6].

The exact process of hypoxia sensing by the type 1 cell of the carotid body (CB) has not yet fully been elucidated, and multiple hypothesis have been proposed. (see Kumar et al. [6] and refs cited herein). Simplified, the reaction of the type 1 cell to hypoxia and the following steps in signal transduction from the carotid body to the respiratory centers in the central nervous system are as follows: hypoxia leads to closure of K^+^-channels, inducing membrane depolarization, which in turn results in the opening of voltage gated Ca^2+^-channels, increasing the intracellular concentration calcium in type 1 CB cells. The increased intracellular Ca^2+^ concentration triggers the release of various neurotransmitters, that eventually result in signal transduction to the afferent carotid sinus nerve, which projects to respiratory centers in the brainstem. The brainstem further coordinates the cardio-respiratory reactions. In practice, a hypoxic stimulus evokes a brisk hyperventilatory response (i.e., the acute hypoxic ventilatory response; AHVR), aimed at increasing the uptake of oxygen through the lungs (and/or removal of arterial carbon dioxide). When hypoxia is sustained for episodes longer than 3 to 5 min, breathing slowly declines from its peak value to reach a new steady state within 20–30 min (i.e., hypoxic ventilatory decline or HVD) [10]. The mechanism of the hypoxic ventilatory decline is multifactorial and related to peripheral chemoreceptor adaptation, an increase in cerebral blood flow (causing a reduction in stimulation of the central chemoreceptors by CO_2_ and other acidic substances) and the central accumulation of inhibitory neuromodulators during sustained hypoxic exposure (such as gamma-amino butyric acid, lactic acid, nitric oxide or platelet-derived growth factor isoforms AA) [10]. In clinical practice, and for the purpose of this review, the discussion is limited to the acute hypoxic ventilatory response. The effect that muscle relaxants and the intravenous hypnotic propofol have on the acute hypoxic ventilatory response will be discussed below.

## 3. The Role of Cholinergic Chemotransmission in Chemotransduction

Throughout the human body, cholinergic neurotransmission plays a major role in multiple autonomous functions at various organ systems; we will primarily focus on the role of the nicotinic acetylcholine receptors (nAChr). For a comprehensive review of the structure and function of nAChr in mammals, we refer to the review of Albuquerque et al. [11]; however, we will provide a brief overview here for general understanding. The nAChr are members of a receptor superfamily of receptor ion-channels that include GABA, 5-HT3 and ATP receptors. Structurally, they consist of 5 subunits, which together form a transmembrane channel, through which, when opened, Na^+^ and Ca^2+^ can flow into the cell, resulting in depolarization of the cell membrane. nAChr are built of combinations of alfa subunits (α1-7 and α9-10, notably α8 subunits are not found in mammals) and non-alfa subunits (β1-4) [11]. In addition, a distinction is made between muscle type and neuronal type nAChr. Muscle type nAChr are composed of two α1 subunits combined with three non-α subunits (notably β1, δ and ε or γ subunits) and are found at the postsynaptic membrane at the neuromuscular junction. Neuronal nAChr subtypes consist of combinations of α2-10 and β2-4 subunits and are found at various neuronal but also non-neuronal tissues in the body. Knowledge of the subunit composition of the nicotinic acetylcholine receptor is important as the agonistic activation–inactivation kinetics, and the (relative) permeability for ions, depend on the subunit composition. For instance, nAChr composed of the α7 subunit are known to desensitize rapidly and have a distinctly higher Ca^2+^:Na^+^ permeability than nAChr composed of other subunits. In general, the (tissue-)specific expression of nAChr subtypes results in (tissue-) specific actions induced by receptor ligands [11].

At the carotid bodies, type 1 cells release various neurotransmitters in response to hypoxemia. These include, but are not limited to, dopamine, norepinephrine, serotonin, purines, neuropeptides and acetylcholine [6,12]. The resultant excitation or inhibition of the afferent nerve synapse depends on the composition of neurotransmitter discharge by the type 1 cell, as well as loco-regional receptor expression on type 1 and type 2 cells and postsynaptic nerve terminals. In general, a consensus on the exact neurotransmission mechanism in the carotid body, including the relative contribution or relevance of the various neurotransmitters in this process, remains open.

Acetylcholine was among the first candidate neurotransmitters that appeared to be involved in the chemotransduction process [13]. Animal experiments revealed that exogenously applied acetylcholine dose-dependently increased CB sinus nerve discharge activity. This was found in various animal species with the exception of rabbits, where expression of muscarinic acetylcholine receptors mediate an inhibitory effect of acetylcholine [14]. Apart from the contradictive effect in rabbits, the function of acetylcholine as primary excitatory neurotransmitter in chemotransduction in other animals has been debated. Experiments by Reyes and colleagues showed that pharmacological cholinergic antagonism in cats does not eliminate hypoxia induced chemoreflex hyperventilation, nor chemosensory excitation [15]. Similar findings were found in in vitro rat experiments, showing that cholinergic antagonists inhibit nicotine induced, but not hypoxia induced increases in intracellular calcium in type 1 cells or afferent nerve action potentials [16,17]. Still, in a whole carotid body model, the nAChr antagonist vecuronium did depress afferent nerve discharge [18]. These results suggest that acetylcholine may not be the primary excitatory neurotransmitter in the signal transduction process, although it does profoundly affect chemotransduction, possibly through presynaptic pathways. Presynaptic functions of cholinergic receptors are observed at other sites in the human body. For instance, presynaptic nicotinic acetylcholine receptors modulate (i.e., facilitate or inhibit) synaptic signal transmission between neurons in the central nervous system and between neuron and muscle-endplate at the neuromuscular junction [19,20]. In general, activation of presynaptic nAChr mediates a positive feedback loop, by increasing the intracellular [Ca^2+^] of the terminal through inflow of Ca^2+^ through the opened nAChr-channels and the opening of voltage-gated Ca-channels. Increased intracellular [Ca^2+^] stimulates the release of (other) neurotransmitters from the terminal. However, a negative modulatory role for presynaptic nAChr has also been observed, although its significance is less clear [19]. A well-studied positive feedback loop, in which presynaptic nAChr are involved, is signal transmission at the neuromuscular junction (NMJ). At the NMJ, presynaptic (neuronal) nAChr expressed at the nerve terminal amplify the signal transmission between the neuron and the motor end-plate in order to meet the demand for acetylcholine in case of repeated or prolonged stimuli [21]. Blocking the presynaptic nAChr results in failure to meet this increased demand, which is clinically detectable as fading muscle strength when repetitive stimuli are applied (e.g., during train-of-four stimulation) [20,21].

Presynaptic acetylcholine receptors expressed on type 1 cells (also called autoreceptors; i.e., nAChr that are expressed on acetylcholine containing terminals) have been identified in carotid body tissue of various animal species (rat, cat, mouse, see Shirahata et al., 2007 [12] for review). Pre- and postsynaptic nAChr differ with regard to their subunit composition, with, in cat CB, presynaptic nAChr being composed of α3, α4 and β2 subunits [22], whereas postsynaptic nAChr consisted of α7 subunits [23]. In addition to nicotinic acetylcholine receptors, M1 and M2 muscarine acetylcholine autoreceptors are also expressed on type 1 cells which modulate chemotransduction as well, albeit through different intracellular pathways [12]. The compound effect of acetylcholine in the chemotransduction process thus depends on the local expression of muscarinic and nicotinergic receptors on the type 1 cell and afferent nerve terminal, where (in general) activation of presynaptic nicotinic and M1 acetylcholine autoreceptors facilitate neurotransmitter release by the type 1 cell, as opposed to inhibition of this process by stimulation of M2 autoreceptors [12]. In addition, cholinergic autoreceptors are involved in modulation of multiple other pre- and postsynaptic receptor systems. This includes inhibition of TASK-like potassium channels through mAChr stimulation, which are important in the chemosensing process of the type 1 cell [24]. These data on the expression of cholinergic receptors in carotid bodies primarily come from animal studies. Data on human carotid body nAChr subunit expression are scarce, however investigation of human CB gene expression has revealed expression of α3, α7 and β2 subunits, indicating the presence of multiple nAChr subtypes at human CBs [25], alike to what was found in various animal species. Nevertheless, as significant interspecies differences exist, animal findings cannot directly be translated to humans and a full understanding of cholinergic transmission in human chemotransduction is not yet present.

## 4. Anesthetic Drug Effects on Peripheral Chemosensitivity

General anesthesia is a pharmacologically induced state which features loss of consciousness and attenuation of (nociception-evoked) autonomic responses and motor reflexes. Anesthesia-induced loss of consciousness is primarily attributed to drug actions on central GABA-ergic transmission, while immobility, obtained by neuromuscular blocking agents, is the result of a block of the muscle-type nAChr at the neuromuscular junction. Apart from their intended effects, many anesthetic agents also interfere with central and peripheral chemosensitivity, in part due to accessory effects on cholinergic neurotransmission. For instance, the effects of muscle relaxants are not limited to a block of muscle-type nicotinic receptors at the neuromuscular junction, as neuronal nicotinic acetylcholine receptors that are expressed at other organs, including the carotid bodies, are also blocked by these agents. In this respect, it is important to distinguish the two classes of muscle relaxants that are used in clinical practice, as these classes have distinct actions on nAChr. Depolarizing muscle relaxants, of which, in contemporary practice, only succinylcholine is used, only interact with the post-synaptic, muscle type nAChr at the neuromuscular junction. When binding at these receptors, they cause depolarization of the postsynaptic membrane, resulting in visible muscle twitches. Subsequently, the muscle receptor is desensitized for several minutes, resulting in paralysis of the muscle. Non-depolarizing muscle relaxants (e.g., atracurium, vecuronium, pancuronium, rocuronium) also bind to the postsynaptic muscle-type nAChr, but do not induce depolarization. Instead, they block the acetylcholine binding site on the α1 subunit of the nAChr, resulting in a dose dependent paralysis. Seminal observations by the group of Eriksson showed that non-depolarizing muscle relaxants are able to profoundly attenuate the acute hypoxic ventilatory response (AHVR) in human volunteers [4,7,8]. By showing that the response to hypercapnia was not significantly affected, it was hypothesized that muscle relaxants exerted these effects within the peripheral chemoreflex loop at the peripheral chemoreceptors. Similar observations have been made recently for the more commonly used muscle relaxant rocuronium [9]. Together, these studies show that a shallow neuromuscular block (e.g., train-of-four muscle twitch response ratio of 70 to 90%, see ref [26]. for details on the depth of neuromuscular block) reduces chemosensitivity in healthy volunteers by approximately 40 to 50%. In experimental follow-up studies, the group of Eriksson confirmed that muscle relaxants vecuronium and atracurium inhibit nicotine-induced peripheral chemoreceptor responses in equipotent doses, indeed suggesting a selective block of neuronal nAChr at the carotid bodies [27]. Finally, in an in vitro model of xenopus laevis oocytes expressing various nAChr subtypes, they confirmed that non-depolarizing muscle relaxants were not only able to block muscle type α1β1 nAChr, but also concentration dependently blocked various neuronal nAChr subtypes [20]. Together, these experiments strongly suggest that non-depolarizing muscle relaxants inhibit peripheral chemosensitivity through a direct nicotinic block of neuronal nAChr that are expressed at the carotid body. However, as cholinergic transmission is primarily involved in modulatory effects in the chemoreflex loop, the exact effect of muscle relaxants on the type 1 cell and chemotransduction process is not known. One example that supports the hypothesis that muscle relaxants interfere with cholinergic chemotransmission is that, following administration of rocuronium, the AHVR is depressed, an effect that is just partly related to the reduction in muscle strength [9].

Apart from the effects of muscle relaxants at the carotid bodies, negative effects of hypnotic anesthetic agents on peripheral chemosensitivity are also observed. Although hypnotic agents exert their hallmark effects through interactions with GABA receptors, various intravenous agents (including propofol, thiopental, etomidate and ketamine) as well as inhalational agents (including isoflurane, enflurane and halothane) extensively interact with neuronal nAChr subtypes as well [28]. These interactions vary among agents and are nAChr subtype specific. Therefore, the extent to which these interactions put relevant strain on peripheral chemosensitivity also differs among agents. For instance, although both inhalational and intravenous anesthetics selectively and independently depress peripheral chemosensitivity [2,29,30,31], their actions on the type 1 cell differ markedly. Evidence from human and animal experiments show that inhalational anesthetics primarily interfere with background potassium channels, inhibiting depolarization of type 1 cells in response to hypoxia [32]. The magnitude of depression however, differs between agents (i.e., halothane > enflurane > isoflurane > sevoflurane) [33]. In contrast, the commonly used intravenous agent propofol has actions on calcium channels and nAChr on the type 1 cell. Similar to what was found for muscle relaxants, in vitro experiments demonstrate that propofol is able to block various muscle and neuronal subtype nAChr. This effect was noted in concentrations below that needed for general anesthesia [34]. Evidence that cholinergic inhibition by propofol underlies its chemodepressant effects comes from various animal and human observations. For instance, in isolated, whole body rat carotid bodies, propofol depresses nicotine induced carotid body afferent nerve discharge [35]. More recently, O’Donohoe and colleagues confirmed in an isolated type 1 cell model that propofol inhibits the nicotine-induced increase in intracellular [Ca^2+^] in type 1 cells, suggesting a direct effect of propofol on cholinergic autoreceptors [16]. Additionally, propofol inhibited type 1 voltage gated Ca^2+^-channels. Interestingly, no effect of exogenous applied GABA on intracellular [Ca^2+^] was noted, although GABA receptors have been identified on type 1 cells and sinus nerve terminals in animal carotid bodies [36,37]. Finally, no effects of propofol on background potassium channels were noted, the primary target of volatile anesthetics [16].

## 5. Anesthetic Perturbation of Peripheral Chemosensitivity: Consequences for Clinical Practice

For anesthesia caregivers, it is crucial to acknowledge the detrimental effects anesthetic agents have on hypoxic ventilatory control. Especially in the postoperative period, where patients weaned off the ventilator are required to breath adequately, residual effects of anesthetics may cause respiratory compromise and complications. Of all anesthetic agents, the cause–effect relation between postoperative respiratory complications and residual effects of muscle relaxants (i.e., residual neuromuscular block) is especially well documented [38,39,40,41,42]. Importantly, residual neuromuscular block (NMB) not only depresses peripheral chemosensitivity, but is also associated with upper airway collapse and reduced upper esophageal sphincter tone, which may precipitate aspiration of gastric contents [43,44]. These effects are caused by a marked higher sensitivity of the upper airway and upper gastrointestinal tract musculature for shallow levels of residual NMB. Counterintuitively, patients often sustain an adequate ventilatory minute volume, despite significant residual NMB, because ventilatory muscles, including the diaphragm, show faster recovery of NMB than upper airway muscles [45]. The carotid bodies are similarly very sensitive to shallow levels of residual NMB. Therefore, residual NMB renders patients at risk for respiratory compromise through multiple detrimental effects. Unfortunately, incidences of residual NMB in clinical practice are substantial [41,46,47], and these episodes often remain undetected by healthcare personnel as patients commonly do not display symptoms of evident muscle weakness. In addition, recent observations by Broens et al. suggest that, even after full recovery of the muscle relaxation (as measured at the thumb by relaxation monitoring with a train-of-four muscle twitch response ratio of 100%), the peripheral chemoreflex remains depressed in a significant subset of subjects [9]. This suggests that a large proportion of patients remains at risk for hypoxia, even when the muscle relaxation has fully worn off according to clinical monitors. Surprisingly, the use of the rapid-acting and potent reversal agent, sugammadex, did not result in significantly better return to baseline values of the AHVR compared to spontaneous recovery of muscle relaxation. In addition, a large inter-individual variation in the magnitude of AHVR inhibition and the time to full recovery were observed [15]. Which mechanisms play a role in this observation is currently unknown, however among factors, receptor desensitization, and interference of muscle relaxants with the presynaptic modulatory pathways of chemosensitivity, are plausible. The duration of AHVR depression, as well as mechanisms that underlie these observations need further study.

Propofol, like muscle relaxants, has the ability to interact with cholinergic chemotransmission at the carotid bodies. Other intravenous and inhalational anesthetics have also shown to depress peripheral chemosensitivity, albeit primarily via non-cholinergic pathways. Currently, the effects of anesthetic agents and muscle relaxants on peripheral chemosensitivity have only been studied in separate and in highly controlled laboratory settings. However, in clinical practice, lingering effects of multiple agents will be encountered simultaneously in the postoperative period. The combined effect of residual concentrations of anesthetic agents on chemosensitivity in the postoperative period is currently unknown, but potentially synergistic. Clinicians and other health care personnel should strive to minimize the negative effects of these agents on peripheral chemosensitivity in the postoperative period in order to reduce the risk for adverse respiratory events. While a certain degree of postoperative residual sedation is not preventable (due to systemic accumulation of hypnotics and opioids), residual neuromuscular block is preventable. Through the application of neuromuscular monitoring devices and the use of reversal agents, anesthesia caregivers can prevent or treat a residual muscle relaxant effect at the end of anesthesia. The level of residual neuromuscular block that is regarded as acceptable (a train-of-four muscle twitch response ratio of 90% or higher) may, however, need further study, considering the anticipated concurrent negative effects of residual sedation in clinical practice.

In conclusion, we have reviewed the role of cholinergic chemotransmission in the chemotransduction process and mechanisms by which neuromuscular blocking agents and hypnotic agents interfere in this process. To gain a full understanding of the effects of anesthesia on hypoxic ventilatory control, studies that further investigate the complex interaction of anesthetics and neuromuscular blocking agents on peripheral chemosensitivity are urgently needed.

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
