# Peer review of "Cholinergic Chemotransmission and Anesthetic Drug Effects at the Carotid Bodies"

_molecules, 2020, doi:10.3390/molecules25245974_

Round 1
Reviewer 1 Report
In this manuscript entitled "Paralyzing the carotid bodies: a scoping review on the effects of muscle relaxants on peripheral chemosensitivity" the authors make a short review of the effect of non-depolarizing neuromuscular blocking agents, commonly used as a supplement to general anesthesia, on peripheral chemosensitivity, discussing the possible consequences for clinical practice.
In my opinion, the subject is interesting, and in principle the manuscript is easy to read, but when it is analyzed more slowly I find major concerns, which I am going to point out.
In the abstract, lines 12-13, it reads "A vital reflex that depends on cholinergic transmission is the peripheral chemoreflex to hypoxia that originates at the carotid bodies". This sentence is very categorical, partial, and misleading, since in this reflex, the response of chemoreceptor cells to hypoxia leads to the release of many other neurotransmitters in addition to acetylcholine (ATP, Catecholamines, among others), therefore the chemoreceptor reflex only partially depends on cholinergic transmission, and it is not known to what degree, in fact there is literature against cholinergic transmission in carotid body chemosensory activity and ventilatory chemoreflexes (EP Reyes et al. Respir Physiol Neurobiol 2007, DOI: 10.1016/j.resp.2006.07.006; DF Donnelly, J Appl Physiol, 2009, DOI: 10.1152/japplphysiol.00135.2009). Will you please comment on this?
Also in the abstract it says “This manuscript reviews the mechanisms by which NMBAs affect peripheral chemosensitivity… ..” Throughout the review little is said about mechanisms and rather effects are described.
In section 2: I think it should be explained a little better that the central and peripheral chemoreceptors do not respond to the same stimuli. It should be specified, if only, that the central chemoreceptors only respond to hypercapnia and acidosis, while the peripheral ones respond to more stimuli, such as hypoxia, hypercapnia, acidosis, hyperthermia and others.
In line 40: “…… chemoreceptive tissue can be found at the aortic arch and at the carotid bodies,…”
I sujerst: “… .. chemoreceptive tissue can be found at the aortic arch, the aortic bodies, and at the carotid artery bifurcation, the carotid bodies, ……”
In page 2 Figure 1, it is not necessary that it be so big, nor the colors so strong, but more important than this is that the letters are not read, they must be bigger and clear.
In relation to the K+ channels, those that are sensitive to hypoxia, are open at resting membrane potential (in normoxia), and in these condition the K+ flux is out of the cell, in figure 1 the K+ enters to the cell. In hypoxia, these channels close and depolarize the cell. Please change this in figure 1. In the text it is well described (page 3 lines 54-63).
Also and since the authors mention it, on page 3, line 64-66, a minimal explanation should be given about why “when hypoxia is sustained longer than 3 to 5 minutes, respiratory rate slowly declines from its peak value to reach a new steady state within 20 minutes ” although a reference is given.
In section 3: If this is a review of the effect of muscle relaxants on nicotinic receptors in the carotid body, the description of the types of nicotinic receptors in this organ should be somewhat broader and not limited to a small paragraph in the section 3, lines 101-110.
In line 112: ”… .and non-cholinergic receptors (e.g., inhibition of K+-channels). Please explain what type of non-cholinergic receptors can produce inhibition of wat type of K+-channels.
Line 115-117: ”… ..Thus, apart …… ..neurotransmission” Please explain this sentence. Cholinergic neurotransmission, being probably important in the chemotransmission between type 1 cells and afferent nerve terminals, acetylcholine is not the unique neurotransmitter in carotid body, therefore it may not be pivotal.
What other (modulatory) effects in the carotid body depend on cholinergic neurotransmission? Explain please
In Section 4: Lines 145: “Together, these experiments strongly suggest that non-depolarizing muscle relaxants inhibit peripheral chemosensitivity through a direct nicotinic block of neuronal nAChr that are expressed in the carotid body”. This sentence is too categorical, in the sense that, even taking into account the experiments described in references 15, 16, and 17, which support it, there is also a bibliography against cholinergic transmission in carotid body chemosensory activity and ventilatory chemoreflexes (EP Reyes et al. Respir Physiol Neurobiol 2007, DOI: 10.1016/j.resp.2006.07.006; DF Donnelly, J Appl Physiol, 2009, DOI: 10.1152/japplphysiol.00135.2009).
In any case, if a clear relationship between shallow residual muscle relaxation and adverse respiratory events in the postoperative period has been found in patients, it is clinically difficult to demonstrate full recovery of the muscle relaxation, and this could be the cause of adverse respiratory events in the postoperative period, and not the effect of muscle relaxants on carotid body cholinergic transmission which is not the only and probably not the main transduction mechanism in the carotid body. Comment please
Reviewer 2 Report
Dear Authors,
Thank you for your paper. I read the manuscript titled: Paralyzing the carotid bodies: a scoping review on the effects of muscle relaxants on peripheral chemosensitivity.
The article is interesting, but it does not bring anything new. The links between the topics you choose do not express a novelty (not suitable for a scientific article), it is rather a book chapter.
BR,
Round 2
Reviewer 1 Report
The authors have made a great effort in reviewing the manuscript, and answering the questions and requirements to improve it.
But there are still some aspects to review.
- This review is not about "cholinergic chemotransduction at the carotid body" but rather about "cholinergic chemotransmision" so I think the title should be changed.
- In section 3. Nicotinic acetylcholine receptors in carotid body. The authors do not describe the types of receptors in the carotid body in these section 3, they do so later in the next section, which is incorrectly numbered “section 44. The role of cholinergic neurotransmission in the chemotransduction process. Also here I think "chemotransduction" should be replaced by "chemotransmission". Please merge these two sections.
- Also section 6 and 7 should be merged. The titles of these sections deal with the same aspects.
- In L206-207 of the manuscript with revisions it reads “This includes inhibition of TASK-like potassium channels through mAChr stimulation, which are important in the chemosensing process of the type 1 cell. [4]". Reference [4] says nothing regarding inhibition of TASK-like potassium channels through mAChr stimulation. Another reference is needed for this statement.
- Authors should still carefully check the text as there is some spelling error.
Reviewer 2 Report
Thank you
Author Response
Thank you